# LDHB suppresses the PDCoV proliferation by targeting viral nucleocapsid protein for autophagic degradation

Xiaohan Wu,[1,2] Shijin Lan,[2] Ying Wang,[2] Shixing Yang,[1,2] Quan Shen,[2] Xiaochun Wang,[2] Yuwei Liu,[2] Hongfeng Yang,[1] Likai Ji,[1,2] Wen Zhang[1,2]

**ABSTRACT**    Porcine deltacoronavirus (PDCoV) is a newly identified enteric coronavirus that causes serious diarrhea and vomiting in pigs, leading to substantial economic losses globally. Studying the molecular interactions between virus and host proteins is crucial for developing new anti-PDCoV strategies. Here, the role and mechanism of lactate dehydrogenase B (LDHB) in PDCoV replication were investigated. LDHB suppresses PDCoV replication in a dose-dependent manner, whereas the knockdown of LDHB via RNA interference enhances virus proliferation in LLC-PK1 cells. Mechanistically, LDHB directly interacts with PDCoV N protein in the cytoplasm. LDHB mediated the autophagic degradation of PDCoV N protein, thereby inhibiting viral replication. To our interests, PDCoV infection or PDCoV N protein expression significantly reduces LDHB expression in cells. Further studies showed that PDCoV N protein, dependent on its LIR motif, binds to the LC3. It facilitates LDHB degradation, possibly as a strategy for viral evasion from host cell cytosolic defense mechanisms. Overall, the present study provided a novel regulatory mechanism of LDHB in PDCoV infection and suggested new avenues for the antiviral strategy.

**IMPORTANCE** This study elucidates the intricate interaction between the PDCoV N protein and LDHB within the context of viral infection and immune evasion strategies. By demonstrating that LDHB can suppress PDCoV replication through a novel mechanism involving the autophagic degradation of the viral N protein, the research highlights the potential of targeting such interactions for antiviral strategies. The findings not only expand our understanding of how PDCoV manipulates host cell pathways to its advantage but also open up new avenues for therapeutic interventions that could mitigate the impact of this and similar viral pathogens.

**KEYWORDS**    LDHB, PDCoV N protein, autophagy, LIR motif

Porcine deltacoronavirus (PDCoV), belonging to the *Deltacoronavirus* genus of the *Coronaviridae* family, is a newly identified enteric pathogenic coronavirus in pigs. PDCoV can cause severe dehydration, vomiting, and severe watery diarrhea leading to death in piglets, thereby causing significant economic losses to the pig industry (1). PDCoV was first identified in pig populations from Hong Kong in 2012 and had its first outbreak in the United States in February 2014 (2, 3). Subsequently, PDCoV has been detected in several countries, including South Korea, Japan, and China. PDCoV is capable of infecting not only pigs but also a variety of other animals, including calves and poultry, demonstrating its potential for cross-host species transmission (4). A 2021 study reported independent infections of PDCoV in children from Haiti, highlighting its zoonotic potential as an emerging pathogen (5). As a novel pathogen, the pathophysiological mechanisms of PDCoV and its interactions with host cells are not well understood. Currently, effective treatments and vaccines against PDCoV infection remain limited.

Address correspondence to Hongfeng Yang, feng102220@163.com, Likai Ji, jilikai01@ujs.edu.cn, or Wen Zhang, zhangwen@ujs.edu.cn.

The authors declare no conflict of interest.

The genome of PDCoV is a single-stranded positive-sense RNA, approximately 25.4 kilobases (6). Its genome includes 5'-untranslated region (UTR), open reading frames (ORF) 1a and 1b, spike (S) protein, envelope (E) protein, membrane (M) protein, nucleocapsid (N) protein, and 3′-UTR (7). The 5' and 3' UTRs play crucial roles in the replication and transcriptional regulation of the viral RNA (8). ORF1a and ORF1b encode a polyprotein precursor comprising various non-structural proteins (nsp1 to nsp16) (9). The S and E proteins of PDCoV are essential for the virus entry, assembly, and release processes (10). The N protein as the major structural component of PDCoV is involved in viral replication, assembly, and immune regulation and plays key roles in the viral life cycle (11).

Lactate dehydrogenase (LDH) is a key glycolytic enzyme located in the mitochondrial membrane, capable of catalyzing the reversible conversion between pyruvate and lactate during glycolysis (12). LDH is composed of two types of subunits, LDHA and LDHB, which are encoded by *LDHA* and *LDHB* genes, respectively (13). LDHB exhibits a high affinity for lactate, preferentially catalyzing the conversion of lactate to pyruvate under oxygen-rich conditions (14–16). LDHB is considered a biomarker for various cancers. LDHB expression is silenced by hypermethylation of the LDHB promoter leading to a shift in the glycolytic pathway in pancreatic cancer cells (17). Suppression of LDHB expression enhances the proliferation, invasion, and migration of pancreatic cancer cells under hypoxic conditions (18). LDHB is specifically upregulated in basal-like/triple-negative breast cancer cell lines and tumors (19). Importantly, recent studies have shown that LDHB is involved in regulating the proliferation of several viruses. CSFV infection in PK-15 cells leads to lactate accumulation mediated by LDHB; overexpression of LDHB inhibits CSFV proliferation, whereas interference with LDHB promotes CSFV replication (20). The Spike S1 domain of SARS-CoV-2 was confirmed to interact with LDHB and inhibit the catalytic activity of LDHB, playing a broader role in altering host cell metabolism, thereby contributing to the pathophysiology of viral infection (21). However, the interaction between LDHB and other viral infections remains unclear.

Autophagy is a highly conserved cellular degradation and recycling process in eukaryotes. Moreover, autophagy plays a crucial role in antiviral immunity, inflammation, and anti-aging processes via maintaining cellular homeostasis (22, 23). In PEDV infection, FUBP3 and BST2 recruited the host E3 ubiquitin ligase MARCH8 to catalyze the polyubiquitination of viral N protein, which is then recognized by NDP52 for autophagic degradation (24, 25). IBDV VP2 protein is ubiquitinated and subsequently degraded via the p62-dependent autophagy pathway, preventing the viral proliferation in cells (26). The capsid protein of SINV is selectively targeted for autophagic degradation via interacting with p62 and LC3 (27). For HSV-1, the autophagy receptor OPTN mediates the degradation of essential viral envelope proteins such as VP16 and gB through autophagy (28). Autophagy can also be exploited by viruses to evade surveillance from host defenses. CSFV infection mediated the LDHB decreasing to induce the mitophagy, blocking the RLR antiviral pathway (20). PDCoV could induce a complete autophagy process by activating the p38 signaling pathway to facilitate virus replication (29, 30). PDCoV could be suppressed by the pharmacological inhibitor of autophagy, such as wortmannin and ergosterol peroxide (EP) in vitro (29). PGAM5 suppressed the replication of PDCoV via mediating the p62-dependent autophagic degradation of PDCoV N protein (31). However, how PDCoV interacts with the host to regulate autophagy is still unknown, and more research is still needed.

In the present study, LDHB was newly identified as having an inhibitory effect on the replication of PDCoV. Mechanically, LDHB directly interacts with the PDCoV N protein and mediates its degradation via the autophagy-lysosomal pathway, which depends on the LC3-interacting region (LIR) motif of the PDCoV N protein but is independent of its ubiquitination.

## RESULTS

### LDHB inhibits PDCoV replication in LLC-PK1 cells

To further investigate the functional mechanism of LDHB during PDCoV infection, we transfected LLC-PK1 cells with a constructed LDHB overexpression plasmid (3.1Flag-LDHB). After 24 h of transfection, the cells were infected with PDCoV at a multiplicity of infection (MOI = 1). At 16 and 28 h post-infection, the expression levels of PDCoV N protein and mRNA were analyzed using western blot and qRT-PCR. The results showed that compared with the 3.1Flag-Vector control group, overexpression of LDHB significantly inhibited the expression of PDCoV N protein and mRNA (Fig. 1A and B). Moreover, this inhibitory effect exhibited a dose-dependent decrease as LDHB levels increased (Fig. 1C and D). These results clearly indicated that LDHB exerts a significant inhibitory effect on PDCoV replication in LLC-PK1 cells.

To further elucidate the antiviral role of LDHB, we transfected LLC-PK1 cells with LDHB-targeting siRNA (siLDHB) to selectively reduce LDHB expression. Protein samples were harvested 24 h post-transfection for subsequent analysis, demonstrating the efficiency of the knockdown (Fig. 1E). After 24 h of transfection, the cells were infected with PDCoV at MOI of 1, and the samples were collected at 16 and 28 h post-infection. qRT-PCR results showed that LDHB expression was significantly downregulated in the siLDHB-transfected cells (Fig. 1F). The expression levels of PDCoV N protein and mRNA were detected using western blot and qRT-PCR. The results demonstrated that compared to the negative control group (siNC), knockdown of LDHB significantly promoted PDCoV replication at both protein and mRNA levels (Fig. 1G and H). These data indicate that silencing LDHB effectively enhances PDCoV replication in LLC-PK1 cells, further establishing LDHB as a potential antiviral host factor.

### LDHB degrades PDCoV N protein via the autophagy-lysosome pathway

Host antiviral proteins bind to viral proteins and regulate their expression in cells, which is an important antiviral mechanism of the host. Based on this theory, several PDCoV-encoded proteins were screened for their potential interaction with LDHB by co-immunoprecipitation (Co-IP) assay. To investigate whether a direct interaction exists between PDCoV N and LDHB, we performed Co-IP assays by co-transfecting HEK-293T cells with the 3.1-Flag-LDHB and 3.1-HA-PDCoV N plasmids. After 24 h, the cell samples were collected, and Co-IP analysis confirmed the interaction between LDHB and PDCoV N proteins (Fig. 2A). To further validate this interaction under more physiological conditions, we infected LLC-PK1 cells with PDCoV and assessed the interaction between endogenous LDHB and PDCoV N proteins, thereby confirming their association in a disease-relevant model. The results suggest that LDHB and PDCoV N proteins could form immunoprecipitation (Fig. 2B). Additionally, GST pull-down assays further validated the direct association between these two proteins (Fig. 2C). To confirm the spatial association between LDHB and PDCoV N proteins within the host cells, indirect immunofluorescence assays were performed. The results revealed the cytoplasm co-localization of LDHB and PDCoV N proteins, indicating that they reside within the same cellular compartment (Fig. 2D). These findings indicate that LDHB interacts with PDCoV N protein in the cytoplasm to influence viral replication and cellular dynamics.

HEK-293T cells were co-transfected with Flag-LDHB and HA-PDCoV N plasmids. We observed that the increased expression of LDHB led to a dose-dependent decrease in the abundance of PDCoV N protein (Fig. 2E). Protein degradation in eukaryotic cells primarily occurs through the autophagy-lysosome pathway and the ubiquitin-proteasome pathway (32). To delineate the pathway through which LDHB mediates the degradation of PDCoV N protein, we treated cells with ubiquitin-proteasome inhibitor (MG132) and autophagy-lysosome inhibitors (NH4Cl, CQ, and 3-MA). These results showed that autophagy-lysosome inhibitors significantly restored N protein expression, whereas MG132 had no significant effect (Fig. 2F). To further substantiate these observations, we replicated the experiment in LLC-PK1 cells by transfecting them with the LDHB plasmid,

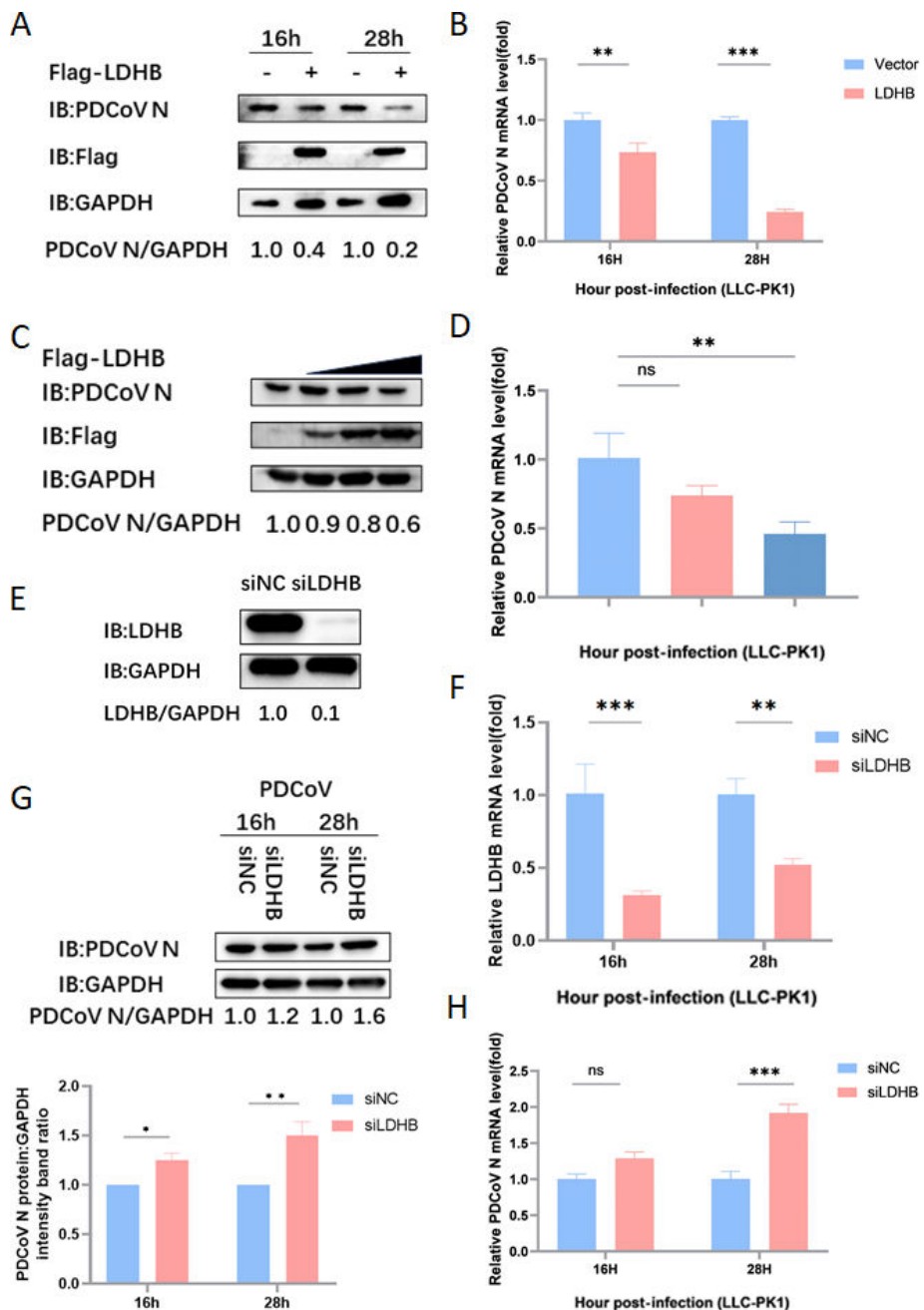

FIG 1  LDHB inhibits PDCoV replication. (A) The LDHB plasmid was overexpressed in LLC-PK1 cells. When the cells reached the appropriate density, they were infected with 1 MOI of PDCoV. Protein samples were gathered at 16 and 28 h for western blot analysis. (B) The mRNA level of PDCoV N was determined through qRT-PCR using the samples described in (A). (C) LLC-PK1 cells were transfected with gradient concentrations of the LDHB plasmids and subsequently infected with 1 MOI of PDCoV. Protein samples were gathered for western blot analysis. (D) The mRNA level of PDCoV N was determined through qRT-PCR using the samples described in (C). (E) The interference efficiency of LDHB siRNA was determined by western blot analysis 24 h after transfection of LLC-PK1 cells. (F) LLC-PK1 cells were transfected with LDHB siRNA and siNC and later infected with 1 MOI of PDCoV. Protein samples were gathered at 16 and 28 h for western blot analysis. qRT-PCR assay was performed to detect the expression of LDHB mRNA. (G) The protein levels were assessed by densitometry of western blots (up). These results were statistically analyzed (down). (H) The mRNA level of PDCoV N was determined through qRT-PCR using the samples described in (F).

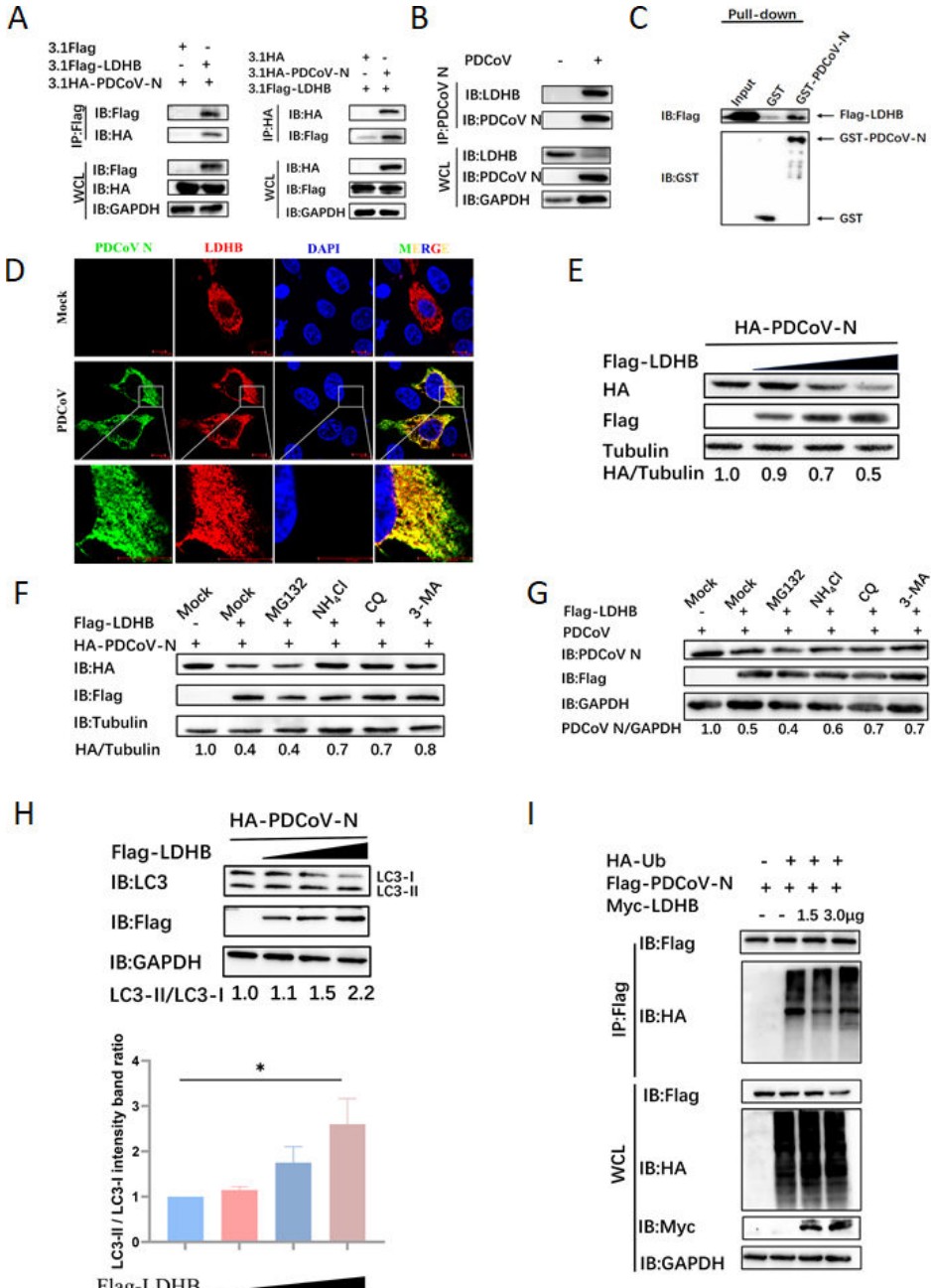

**FIG 2** LDHB interacts with the N protein and degrades it via autophagy. (A) HEK-293T cells were co-transfected with Flag-LDHB and HA-PDCoV N plasmids, and cell samples were collected for Co-IP analysis. (B) In LLC-PK1 cells infected with PDCoV, the endogenous interaction between LDHB and N protein was detected using immunoprecipitation (IP) assays. (C) The PDCoV N protein was inserted into the pCold-GST expression vector and then expressed in BL21 cells. The interaction between LDHB and the N protein was explored using a GST pull-down assay. (D) LDHB and PDCoV N plasmids were transfected into PK15 cells, followed by incubation with labeled antibodies. The co-localization of these proteins was then observed using laser confocal microscopy. (E) HEK-293T cells underwent transfection with plasmids encoding LDHB and PDCoV N, followed by the collection of protein lysates for subsequent analysis using western blotting. (F) After co-transfection with LDHB and PDCoV N plasmids, the cells were treated with ubiquitin-proteasome inhibitors (MG132: 10 µM) and autophagy-lysosome inhibitors (NH4Cl: 12.5 µM, CQ: 50 µM, 3-MA: 0.5 mM) for 8 h to assess the pathways involved in protein degradation. (G) After infecting LLC-PK1 cells with PDCoV (MOI = 1) for 24 h, the cells were treated with the same inhibitor as described above to evaluate the effects of these inhibitors on viral protein degradation and replication. (H) HEK-293T cells were transfected with plasmids carrying LDHB and PDCoV N, and protein extracts were subsequently harvested for western blot analysis. The

Fig 2 (Continued)

blots were incubated with LC3 antibodies to assess changes in LC3 expression (up). These results were statistically analyzed (down). (I) HEK-293T cells were co-transfected with plasmids encoding PDCoV N, Ubiquitin (Ub), and LDHB. Protein samples were subsequently collected for Co-IP assays. Specific antibodies were then used for western blotting analysis to examine the interactions among these proteins

followed by PDCoV infection. In line with our previous results, treatment with autophagy-lysosome pathway inhibitors (NH4Cl, CQ, and 3-MA) partially restored N protein levels, whereas the ubiquitin-proteasome pathway inhibitor (MG132) did not exhibit a noticeable effect (Fig. 2G). Furthermore, LDHB overexpression significantly promoted N protein-induced LC3-II formation (Fig. 2H), indicating that LDHB mediates PDCoV N protein degradation via the autophagy-lysosome pathway, thereby inhibiting viral replication. In most cases, substrate proteins are recognized by autophagy receptors following ubiquitination, initiating autophagy. To investigate the impact of LDHB on the ubiquitination of PDCoV N protein, we transfected PK15 cells with plasmids encoding PDCoV N, ubiquitin (Ub), and LDHB and collected protein samples 24 h later for Co-IP assay. The results showed that LDHB did not affect the polyubiquitination of the PDCoV N protein (Fig. 2I). Hence, LDHB might degrade PDCoV N protein through a non-ubiquitin-mediated autophagy-lysosomal pathway.

## PDCoV downregulates LDHB via N protein-mediated autophagy pathway to evade host antiviral defense

To further investigate the potential role of LDHB during PDCoV infection, we first analyzed the regulatory effect of PDCoV infection on LDHB expression in LLC-PK1 cells. Using qRT-PCR and western blot to measure LDHB mRNA and protein levels after PDCoV infection, the results indicated that LDHB protein levels were marginally affected at 12 and 24 h post-infection but exhibited a notable decline by 36 h post-infection (Fig. 3A). Notably, LDHB mRNA levels remained unchanged at all time points, indicating that PDCoV infection did not significantly affect LDHB transcription (Fig. 3B) but instead promoted protein degradation, leading to the reduction of LDHB protein levels. This suggests that PDCoV may downregulate LDHB through specific protein degradation pathways as part of its strategy to evade the host immune system.

To further explore the role of the PDCoV N protein in this process, we co-transfected HEK-293T cells with 3.1Flag-LDHB and 3.1HA-PDCoV N plasmids. Our findings show that as the expression level of the N protein increases, the abundance of LDHB decreases in a dose-dependent manner (Fig. 3C). To ascertain the specific pathway through which N protein mediates LDHB degradation, we treated the cells with ubiquitin-proteasome inhibitors (MG132) and autophagy-lysosome inhibitors (NH4Cl, CQ, and 3-MA). The results indicate that autophagy-lysosome inhibitors significantly restored LDHB expression, whereas MG132 did not have a noticeable effect (Fig. 3D). It is suggested that PDCoV N protein could mediate LDHB degradation through an autophagy-lysosomal pathway. When investigating the impact of PDCoV N on the ubiquitination of LDHB protein, we found that PDCoV N did not significantly affect the ubiquitination of LDHB (Fig. 3E). It indicated that PDCoV N protein may degrade LDHB protein through a non-ubiquitin-mediated autophagy-lysosomal pathway.

## The role of the LIR domain in mediating PDCoV N protein and LDHB interaction and autophagy regulation

Autophagy receptors commonly contain a conserved LC3-interacting region (LIR) motif that specifically binds to ATG8 family proteins within autophagosomes (e.g., LC3), initiating the protein degradation process (33). In our study, we predicted and validated that the PDCoV N protein has a potential LIR motif, ASWFQV, that binds LC3 (Fig. 4A). Using upright fluorescence microscopy, we demonstrated that PDCoV nucleocapsid (N) protein significantly induced LC3 puncta formation in PK-15 cells, whereas no such

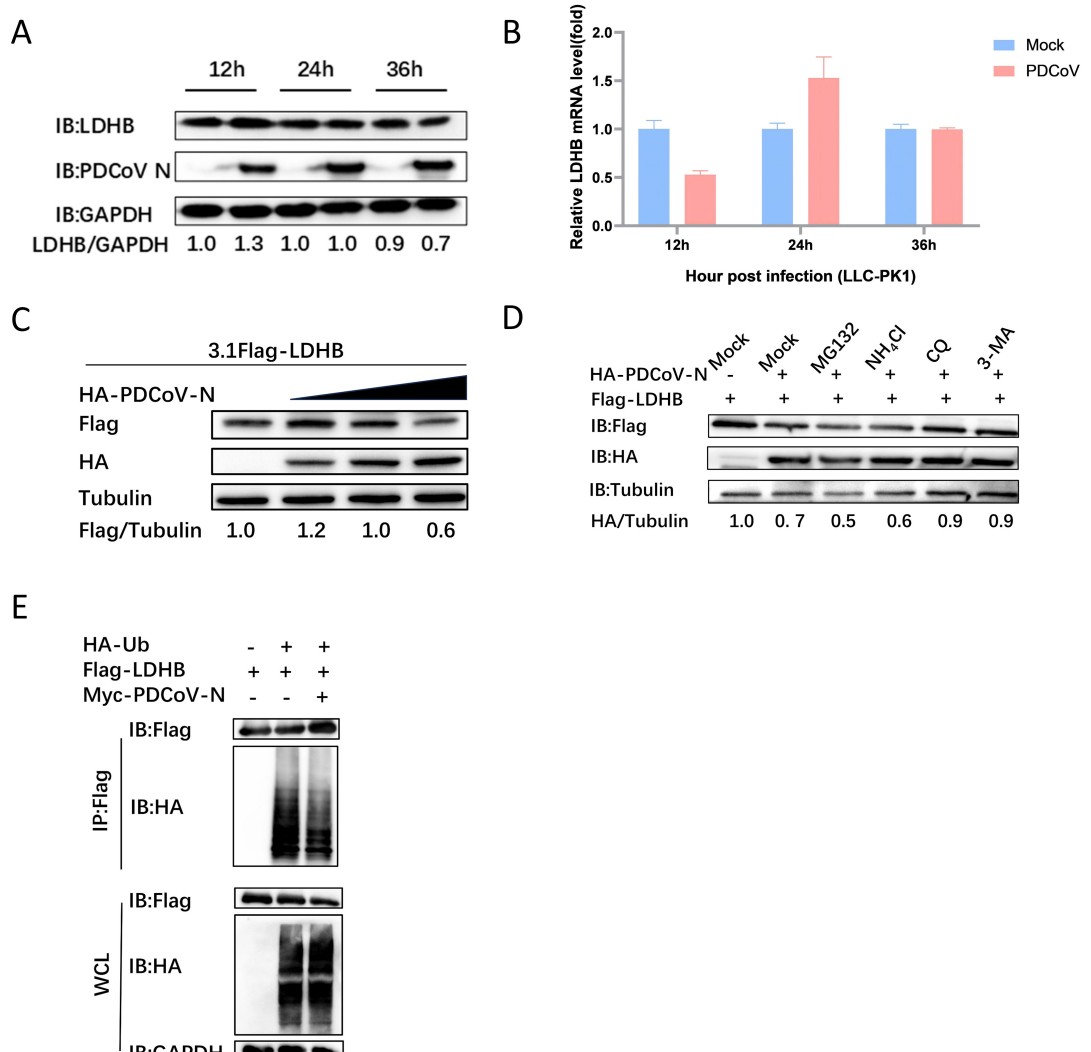

**FIG 3** N protein interacts with LDHB and degrades it via autophagy. (A) After PDCoV infects LLC-PK1 cells, protein samples were collected at 12 h, 24 h, and 36 h post-infection, and the intracellular levels of LDHB were analyzed through western blot assays. (B) The mRNA level of LDHB was determined through qRT-PCR using the samples described in (A). (C) HEK-293T cells underwent transfection with plasmids encoding LDHB and PDCoV N, followed by the collection of protein lysates for subsequent analysis using western blotting. (D) After co-transfection with LDHB and PDCoV N plasmids, the cells were treated with ubiquitin-proteasome inhibitors (MG132: 10 µM) and autophagy-lysosome inhibitors (NH4Cl: 12.5 µM, CQ: 50 µM, and 3-MA: 0.5 mM) for 8 h to assess the pathways involved in protein degradation. (E) HEK-293T cells were co-transfected with plasmids encoding PDCoV N, Ub, and LDHB. Protein samples were subsequently collected for Co-IP assays. Specific antibodies were then used for western blotting analysis to examine the interactions among these proteins.

aggregation was observed in the control group (Fig. 4B, up). Further validation via laser scanning confocal microscopy revealed analogous LC3 redistribution in LLC-PK1 cells, with pronounced co-localization between LC3-positive autophagosomes and PDCoV N protein (Fig. 4B, down). Furthermore, the interaction of PDCoV N protein with LC3 was confirmed in HEK-293T cells by the Co-IP assay with GFP-tagged LC3 (GFP-LC3) or endogenous LC3, respectively (Fig. 4C–E). These results indicated that the PDCoV N protein could interact with LC3 in the cytoplasm. Moreover, we co-transfected Flag-LDHB, GFP-LC3, and HA-N protein plasmids to explore the effects of escalating N protein levels on LDHB stability and its interaction dynamics with LC3. The results demonstrate that increased expression of N protein markedly accelerates the degradation of LDHB in GFP-LC3-expressing cells (Fig. 4F). This reduction in LDHB availability results in decreased interaction signals between LDHB and LC3 (Fig. 4F). These results suggest that PDCoV

N promotes LDHB degradation by impacting autophagic flux through LC3 interaction mediated through its LIR motif.

To further explore the function of the LIR motif in the PDCoV N protein, we constructed a deletion mutant of this motif (PDCoV NΔLIR) and co-expressed it with GFP-LC3 in HEK-293T cells. Co-IP assay showed that deletion of the LIR motif of the PDCoV N protein significantly blocked its interaction with LC3 (Fig. 4E). To explore the role of the LIR motif in the binding of LDHB to the PDCoV N protein during autophagic degradation, PDCoV NΔLIR and LDHB plasmids were co-transfected into HEK-293T cells. The results showed that LDHB could not mediate the degradation of PDCoV NΔLIR protein (Fig. 4G). Moreover, LDHB was also stably expressed in the PDCoV NΔLIR-expressing cells (Fig. 4H). These findings indicate that the autophagic degradation between LDHB and PDCoV N protein is dependent on the LIR motif of the PDCoV N protein, which acts as a viral autophagy receptor to promote the formation of autophagosomes.

## DISCUSSION

PDCoV, as an emerging porcine enteropathogenic coronavirus, causes acute diarrhea and vomiting in neonatal piglets and poses a potential risk for cross-species transmission (34). Although LDHB is more commonly studied in cancer and metabolism-related research, its role in viral infection and replication has begun to receive attention. For example, silencing LDHB can activate NF-κB signaling through mitophagy, promoting the replication of classical swine fever virus (CSFV) (20). Here, we confirmed that LDHB significantly inhibited PDCoV proliferation in a dose-dependent manner, whereas knockdown of LDHB expression via RNA interference enhanced viral replication in LLC-PK1 cells (Fig. 1). These findings indicate that LDHB is a potent antiviral factor against PDCoV infection.

The ubiquitin-proteasome pathway and the autophagy-lysosome pathway are the primary mechanisms of host-mediated viral protein degradation, thereby inhibiting viral replication and spread (35). In the ubiquitin-proteasome system, proteins tagged with ubiquitin undergo the sequential addition of multiple ubiquitin molecules via a series of enzymes, forming monoubiquitination or polyubiquitination chains. This tagging directs proteins to the 26S proteasome for recognition and degradation (36). For example, HDAC6 targets PDCoV-encoded non-structural protein 8 (nsp8) by deacetylating lysine 46 (K46) and ubiquitinating K58, leading to its degradation via the ubiquitin-proteasome pathway and thus inhibiting viral replication (37). Autophagy is another critical host defense mechanism, which can be selective, targeting specific viral components, or non-selective, acting as a broader cellular stress response. Autophagy involves the formation of autophagosomes that engulf viral particles or proteins, which then fuse with lysosomes to degrade the enclosed contents. It not only degrades viral proteins but also includes viral genetic material and entire viral particles. For instance, studies have shown that PGAM5 interacts with the cargo receptor p62 and the E3 ubiquitin ligase STUB1 to degrade PDCoV N protein via the host autophagy pathway and induces interferon production through the MyD88-TRAF3-IRF7 pathway, thereby countering PDCoV replication (31). Here, we found that LDHB can directly interact with PDCoV N protein in the cytoplasm, directing it for degradation through the autophagy-lysosome pathway (Fig. 2A through G). LDHB-mediated degradation of PDCoV N protein is dependent on the autophagy-lysosome pathway, but not the ubiquitin-proteasome system (Fig. 2I).

Selective autophagy is a key viral escaping mechanism for degrading host antiviral factors. In the present study, PDCoV infection and N protein expression significantly reduced LDHB protein levels without affecting its mRNA expression (Fig. 3). It indicated that PDCoV appears to counteract this antiviral mechanism by downregulating LDHB expression via post-transcriptional regulation. Protein ubiquitination is an important post-translational modification process that marks target proteins for degradation via the proteasome or autophagy (38). However, ubiquitination does not affect both the LDHB and PDCoV N proteins after their interaction (Fig. 2I and 3E). To our interests, PDCoV N protein could directly interact with LC3 through its LIR motif, mediating LDHB

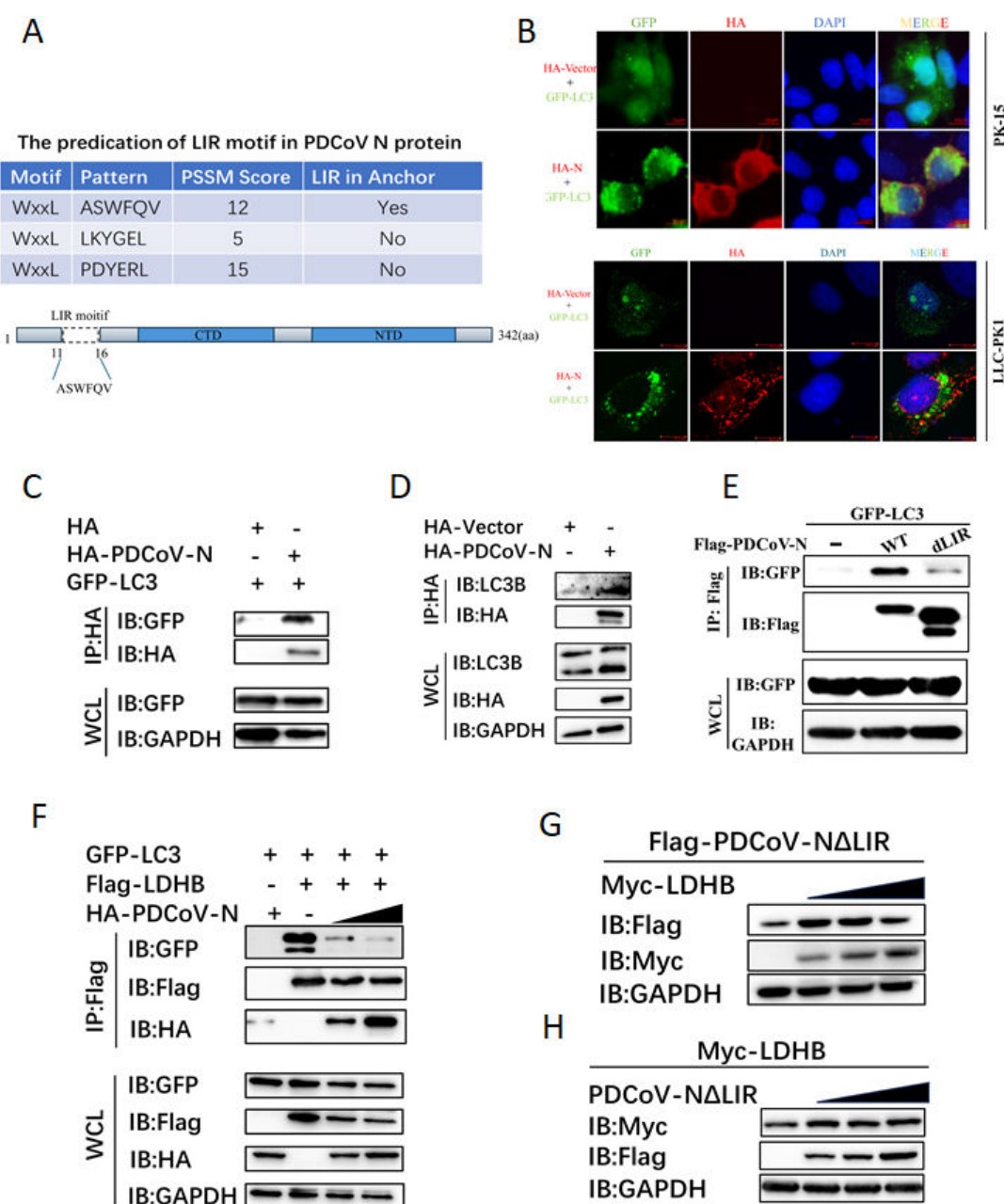

**FIG 4** Functional study of autophagy receptor PDCoV N protein interaction with LDHB. (A) The LIR motif of PDCoV N protein was predicted with Ilir@Viral (https://ilir.warwick.ac.uk/virus/index.php). (B) GFP-LC3 was co-transfected with PDCoV N or empty plasmids, and its aggregation was observed under a fluorescence microscope in PK15 (up) or LLC-PK1 (down) cells. IFA assay was performed to acquire these images under fluorescence microscopy. (C) GFP-LC3 was co-transfected with PDCoV N or empty plasmid, and the aggregation of LC3 in LLC-PK1 cells was observed under laser confocal microscopy. (D) HEK-293T cells were transfected with HA-tagged PDCoV-N. The presence of endogenous LC3 in the precipitates was detected via western blot. (E) GFP-LC3 was co-transfected with Flag-tagged PDCoV-N or NdLIR plasmids into HEK-293T cells for interaction studies. (F) HEK-293T cells were co-transfected with the same dose of LDHB and GFP-LC3 plasmids and increasing amounts of HA-tagged PDCoV N protein. The LDHB-LC3 interaction was assessed by immunoprecipitation and western blot. (G) Overexpressing the LDHB plasmid to observe whether PDCoV NΔLIR is degraded in HEK-293T cells. (H) Overexpressing the PDCoV NΔLIR plasmid to observe whether LDHB is degraded in HEK-293T cells.

autophagic degradation (Fig. 4). We observed that GFP-LC3 fluorescence intensity was reduced, likely due to LC3 fusion with lysosomes. The acidic environment within lysosomes may have quenched GFP fluorescence, providing further evidence of autophagy progression induced by PDCoV N protein. The PDCoV NΔLIR protein, lacking the LIR motif, blocked the interaction with LDHB and prevented LDHB degradation (Fig. 4). This

strategy allows PDCoV to diminish the antiviral effects of LDHB, creating a more favorable environment for viral replication. Specifically, PDCoV N protein accelerates LDHB degradation, significantly reducing the levels of intact LDHB protein in cells. Since the interaction between LDHB and LC3 depends on the availability of LDHB, N protein-mediated LDHB degradation directly leads to a reduction in the formation of LDHB-LC3 complexes, thereby weakening their interaction signals (Fig. 4). This indicates that the N protein not only degrades LDHB to weaken its antiviral function but may also disrupt the LDHB-LC3 interaction to further interfere with the antiviral activity of the host autophagy pathway. The mutual degradation between LDHB and PDCoV N protein highlights the complex interactions between the host and the virus. Although LDHB inhibits viral replication by mediating autophagic degradation of N protein, PDCoV utilizes its N protein to degrade LDHB, thereby evading the host's antiviral response. This tug-of-war underscores the evolutionary adaptation of PDCoV in manipulating host cellular pathways for its own benefit.

In conclusion, our study reveals a novel antiviral mechanism in which LDHB inhibits PDCoV replication by mediating autophagic degradation of the viral N protein. Conversely, PDCoV counteracts this defense by utilizing its N protein, dependent on the LIR motif, to degrade LDHB via the autophagy-lysosome pathway. These findings strengthen our understanding of PDCoV-host interactions and may provide valuable information for developing new antiviral strategies targeting them. However, our study has limitations. The exact molecular mechanisms underlying the interaction between LDHB and PDCoV N protein require further investigation. Additionally, the impact of LDHB downregulation on cellular metabolism and the overall antiviral response during PDCoV infection remains to be fully elucidated. Delving into these questions may help reveal new therapeutic targets.

## MATERIALS AND METHODS

### Antibodies and reagents

We obtained the following antibodies and reagents for our study: LDHB antibody, GAPDH antibody, GST antibody, LC3 antibody, horseradish peroxidase (HRP)-labeled anti-mouse IgG antibody, and HRP-labeled anti-rabbit IgG antibody, all obtained from Abclonal. The PDCoV N protein antibody was prepared by our laboratory. We also used the following reagents: chloroquine phosphate (CQ, Cat. No: PHR1258), 3-methyladenine (3-MA, Cat. No: M9281), MG132 (Cat. No: M7449), anti-MYC-tag antibody (Cat. No: 2276S), and HA-tag antibody (Cat. No: H6908), all acquired from Sigma-Aldrich.

### Cell culture and virus

HEK-293T cells (ATCC) and porcine kidney cells (PK-15 cells, ATCC) were kept in Dulbecco's modified Eagle medium (DMEM), high glucose (Cat.No: E600003-0500, Sangon Biotech) supplemented with 10% fetal bovine serum (Cat.No: F0193, Sigma-Aldrich). LLC-PK1 cells acquired from Tongling Shan (Shanghai Veterinary Research Institute, Shanghai, China) were cultured in minimal essential medium (MEM) (Cat. No: 11095080, Gibco). Both cell lines were subject to incubation at 37°C in a 5% $CO_2$ atmosphere. In the present study, the applied PDCoV strain was isolated from our laboratory.

### Plasmids and transfection

Plasmids were generated using a one-step cloning kit (Cat. No: C112–02, Vazyme Biotech) through homologous recombination. Liposome transfection reagent (Cat. No: C0526, Beyotime) was used for plasmid transfection when cells reached 80%–90% confluency. Interfering RNA was synthesized by Shanghai GenePharma Company and transfected using Lipo6000™ transfection reagent (Cat. No: C0526, Beyotime).

## qRT-PCR

Total RNA was extracted from cells using TRIzol (Cat. No: 15596026CN, Thermo Fisher Scientific). Following extraction, we reverse-transcribed the total RNA into cDNA with the HiScript III RT SuperMix for qPCR (+gDNA wiper) (Cat. No: R323–01, Vazyme Biotech). Subsequently, qRT-PCR was performed using the MagicSYBR Mixture (Cat. No: CW3008M, CWBIO). The list of primer sequences employed in qRT-PCR is presented (Table 1).

## Western blot assay

First, the cells were rinsed with cold phosphate-buffered saline (PBS) and then incubated on ice with radioimmunoprecipitation assay (RIPA) lysis buffer (Cat. No: P0013D, Beyotime) supplemented with protease inhibitors (Cat. No: SB-WB016, Share-bio) for 15 min. 5× SDS PAGE sample loading buffer (Cat. No: 20315ES05, YEASEN) was added. Then, the samples were boiled for 15 min. SDS-PAGE was then conducted to separate the proteins. After electrophoresis, the membrane was blocked with 5% skim milk (Cat. No: 36120ES76, YEASEN) for 120 min. The membrane was incubated with the primary antibody at 4°C for 12 h, followed by three washes with Tris-buffered saline with Tween 20 (TBST). The next step involved incubating the membrane with a secondary antibody for 60 min and then washing it again with TBST. Protein detection was finally carried out using an ECL luminescent solution (Cat. No: SB-WB012, Share-bio).

## Co-IP assay

After 24 h of plasmid transfection, cell lysates were collected using RIPA lysis buffer. The supernatant was then added to pre-resuspended Anti-Flag affinity gel (Cat. No: P2282, Beyotime) or Anti-HA affinity gel (Cat. No: P2287, Beyotime) and incubated on a rotating mixer at 4°C for 4 h. After incubation, the beads were washed three times with 1× TBS, and 5× SDS PAGE sample buffer was added to prepare the protein samples. The eluted proteins were subsequently analyzed using western blot.

## GST affinity isolation assay

The gene sequence of PDCoV N was cloned into the pCold-GST vector, and the resulting plasmid was transformed into *E. coli* BL21 cells (Cat. No: B528419-0010, Sangon Biotech). To investigate protein-protein interactions, we performed a GST pull-down assay following the manufacturer's protocol for GST-Sep glutathione agarose resin (Cat. No: 20507ES05, YEASEN). The samples were subsequently analyzed using western blotting.

## Immunofluorescence assay

After 24 h post-transfection, the cells were fixed with 4% paraformaldehyde. The fixed cells were then permeabilized using 0.1% Triton X-100. Blocking was performed with 3% bovine serum albumin (BSA) at 37°C for 1 h. Primary antibodies, diluted in 3% BSA, were incubated with the cells at 37°C for 4 h. After washing three times with PBS, the cells were incubated with secondary antibodies for 1 h. Finally, 4',6-diamidino-2-phenylindole (DAPI) staining solution, prepared in PBS, was applied to stain the nuclei. Samples were observed using a confocal immunofluorescence microscope (Leica TCS SP5).

## Statistical analysis

Data from three independent experiments were expressed as means ± standard deviations. Significance was determined with a two-tailed Student's t test to analyze the differences in multiple groups (>=3). $P$ values of $< 0.05$ were considered statistically significant.

## ACKNOWLEDGMENTS

This research was supported by National Key Research and Development Programs of China (No. 2023YFD1801301 and 2022YFC2603801), the National Natural Science

**TABLE 1** Sequences of the primers and siRNAs

| Purpose and primer name | Sequence (5'–3') |
|---|---|
| PCR primers | |
| 31 Flagg-LDHB forward | CTAGCGTTTAAACTTAAGCTTATGAGGGGCGGGCCC |
| 31 Flagg-LDHB reverse | TGCTGGATATCTGCAGAATTCGACAGGTCCTTCAGATC |
| HA-PDCoV NΔLIR-1 forward | GTCTTGGTTTCAGGTGCTCA |
| HA-PDCoV NΔLIR-2 forward | ATGGCCGCACCAGTAGTCCCTACTGTCTTGGTTTCAGGTGCTCA |
| HA-PDCoV NΔLIR-3 forward | GACGACGATGACAAGGGTACCATGGCCGCACCAGTAGTCC |
| HA-PDCoV NΔLIR reverse | AAAAAGATCTGCTAGCTCGAGCTACGCTGCTGATTCCTGCTTT |
| siRNA sequences | |
| si-LDHB sense | CCUGGAAGUAAGUGGAUUTT |
| si-LDHB antisense | AAUCCACUUAGCUUCCAGGTT |
| NC sense | UUCUCCGAACGUGUCACGUTT |
| NC antisense | ACGUGACACGUUCGGAGAATT |

Foundation of China (No. 82341106 and 32102682), Medical Education Collaborative Innovation Fund of Jiangsu University (No. JDY2022008), and the Postdoctoral Science Foundation of China (grant No. 2022M721391).

W.Z., L.J., and H.Y. conceived and designed the experiments. X.W., S.L., Y.W., and S.Y. performed the experiments and analyzed data. L.J., Y.L., X.W., and Q.S. wrote the manuscript. All authors read the manuscript and agreed to its contents.

## AUTHOR AFFILIATIONS

[1]Institute of Critical Care Medicine, The Affiliated People's Hospital, Jiangsu University, Zhenjiang, China
[2]School of Medicine, Jiangsu University, Zhenjiang, China

## AUTHOR ORCIDs

Xiaohan Wu  http://orcid.org/0009-0003-2501-1195
Quan Shen  http://orcid.org/0000-0001-9041-8475
Yuwei Liu  https://orcid.org/0000-0002-5495-2926
Hongfeng Yang  http://orcid.org/0000-0001-9624-6109
Likai Ji  http://orcid.org/0009-0007-0411-7774
Wen Zhang  http://orcid.org/0000-0002-9352-6153

## AUTHOR CONTRIBUTIONS

Xiaohan Wu, Data curation, Formal analysis, Writing – original draft | Shijin Lan, Data curation, Formal analysis | Ying Wang, Data curation, Formal analysis | Shixing Yang, Data curation, Formal analysis | Quan Shen, Data curation, Formal analysis | Xiaochun Wang, Data curation, Formal analysis | Yuwei Liu, Data curation, Formal analysis | Likai Ji, Methodology, Supervision | Wen Zhang, Methodology, Supervision.

## ADDITIONAL FILES

The following material is available online.

Open Peer Review

**PEER REVIEW HISTORY (review-history.pdf).** An accounting of the reviewer comments and feedback.

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
