## [Reviewer comments · Microbiology Spectrum]

Microbiology Spectrum

LDHB suppresses the PDCoV proliferation via targeting viral nucleocapsid protein for autophagic degradation

Xiaohan Wu, Shijin Lan, Ying Wang, Shixing Yang, Quan Shen, Xiaochun Wang, Yuwei Liu, hongfeng yang, Likai Ji, and Wen Zhang

Corresponding Author(s): Wen Zhang, Jiangsu University

Review Timeline:

Submission Date:	November 4, 2024
Editorial Decision:	January 8, 2025
Revision Received:	February 28, 2025
Accepted:	March 11, 2025

Editor: Leiliang Zhang

Reviewer(s): The reviewers have opted to remain anonymous.

Transaction Report:

DOI: <https://doi.org/10.1128/spectrum.02787-24>

Re: Spectrum02787-24 (LDHB suppresses the PDCoV proliferation via targeting viral nucleocapsid protein for autophagic degradation)

Dear Prof. Wen Zhang:

Thank you for the privilege of reviewing your work. Below you will find my comments, instructions from the Spectrum editorial office, and the reviewer comments.

Revision Guidelines

Sincerely,
Leiliang Zhang
Editor
Microbiology Spectrum

Reviewer #1 (Comments for the Author):

no

Reviewer #2 (Comments for the Author):

In the manuscript titled "LDHB suppresses the PDCoV proliferation via targeting viral nucleocapsid protein for autophagic degradation," the authors investigate the role of LDHB in PDCoV replication. The study found that LDHB suppresses viral proliferation. The PDCoV-N protein was shown to interact with LDHB. The authors also found an LIR motif within the N-protein. While the study has provided some compelling evidence showing LDHB affecting PDCoV levels, some major corrections to strengthen the paper are suggested below -

1. Throughout the results, multiple cell lines have been used. Please indicate which cell line was used above/near each blot/IF/qPCR.

2. Beyond figure 1, none of the results have been validated in PDCoV infected cells.

3. Fig 1

1E - In the blot confirming the efficacy of LDHB siRNA, at what time points were the samples collected?

Fig 1F - Changes in PDCoV N protein is not very apparent. Plot a bar graph with changes in PDCoV-N/GAPDH over multiple experiments and include other blots in supplemental data. It would be helpful to confirm knockdown efficiency by LDHB western blots at the time points corresponding to 1F.

4. Fig 2 - The data would be compelling if authors could replicate the experiment in 2E (using the various autophagy inhibitors) on PDCoV infected LLC-PK1 and measure viral replication to further validate the role of LDHB in an autophagy-mediated mechanism.

2C - The IF images are too small to see.

2F - Include images with darker/longer exposure to see the LC3-I band. And, include LDHB/Flag blot(s).

2G - Show LDHB/Myc blots.

5. Fig 3A - Authors state the LDHB protein was "significantly" decreased at 24h post infection (line 141) - this decrease is not very apparent. Also, if the authors have done statistics over multiple western blots to determine significance, include the data in the figure.

6. Fig 4 -

4A - What software/resource was used to predict the LIR motif (line 162) - detail in methods.

4B - The IF images are too small to see. It is hard to see the canonical structures formed by the autophagosomes due to the small image size and it is hard to see the co-localization. Include quantification of the autophagosomes with HA-N overlap.

Minor Edits

1. Line 15 - Rephrase "to our interesting"

2. Line 123 - Change "clarify" to delineate.

In this paper, Wu et al. investigated the role of LDHB on PDCoV infection and replication. The authors found that the expression of LDHB was an important antiviral factor against PDCoV infection. Furthermore, the authors also demonstrated that the LDHB could interact with PDCoV N protein, and induce the autophagic degradation of PDCoV N protein. PDCoV N protein depending on its LIR interact with LC3 to degrade LDHB to escaping the host's antiviral mechanism. There are some of issues that should be addressed by the authors.

Major comments:

1. Lines 117-119: The authors point out “PDCoV N protein and LDHB co-localize at the mitochondria”. However, In Fig.2C the authors had proved that the LDHB and PDCoV N protein partially colocalized in the cytoplasm, but without any mitochondria marker presenting. Please provide the supporting data.

2. In the present study, authors only investigated the interaction between overexpressed LDHB and PDCoV N protein in Figure 2, the authors should verify the interaction between endogenous LDHB and PDCoV N protein during viral infection.

3. In the present study, authors only investigated the interaction between overexpressed GFP-LC3 and PDCoV N protein in Figure 5, the authors should verify the interaction between endogenous LC3 and PDCoV N protein during viral infection

4. Since your study focuses on the relationship between LDHB, N protein, and LC3, authors have only shown that N protein can induce the LC3-II increasing in the presence of LDHB. The interaction of LDHB and LC3 should be confirmed with or without PDCoV infection or PDCoV N protein expression, which help clarify the mechanism of LDHB and N protein interacting with LC3 for autophagic degradation. The results should be discussed in the part of Discussion.

Mini comments:

1. Line 15 and 214: The “To our interesting” should be corrected by “To our interests”.

2. Line 69-70: The cited content of the literature was described incorrectly. Please correct it.

3. Lines 168-170: The description of the LIR deletion mutant (N Δ LIR) is incomplete. The specific position and sequence of the LIR motif in the PDCoV N protein sequence needs to be added in the results and Fig.4A.

4. The materials and methods are too simple. Please provide more details. For example, the primer sequences for constructing plasmids should be added in Table 1, such as PDCoV N, LDHB and N- Δ LIR mutant.

5. Lines 406-432: In the legends Fig.2 and Fig.3, the authors should add the details regarding the experimental conditions, such as the concentrations and treatment durations of the inhibitors.

6. Overall, the English language should be improved by consulting native speaker or Author Writing Services.

In the manuscript titled “LDHB suppresses the PDCoV proliferation via targeting viral nucleocapsid protein for autophagic degradation,” the authors investigate the role of LDHB in PDCoV replication. The study found that LDHB suppresses viral proliferation. The PDCoV-N protein was shown to interact with LDHB. The authors also found an LIR motif within the N-protein. While the study has provided some compelling evidence showing LDHB affecting PDCoV levels, some major corrections to strengthen the paper are suggested below –

1. Throughout the results, multiple cell lines have been used. Please indicate which cell line was used above/near each blot/IF/qPCR.
2. Beyond figure 1, none of the results have been validated in PDCoV infected cells.
3. Fig 1
1E – In the blot confirming the efficacy of LDHB siRNA, at what time points were the samples collected?
Fig 1F – Changes in PDCoV N protein is not very apparent. Plot a bar graph with changes in PDCoV-N/GAPDH over multiple experiments and include other blots in supplemental data. It would be helpful to confirm knockdown efficiency by LDHB western blots at the time points corresponding to 1F.
4. Fig 2 – The data would be compelling if authors could replicate the experiment in 2E (using the various autophagy inhibitors) on PDCoV infected LLC-PK1 and measure viral replication to further validate the role of LDHB in an autophagy-mediated mechanism.

2C - The IF images are too small to see.
2F – Include images with darker/longer exposure to see the LC3-I band. And, include LDHB/Flag blot(s).
2G - Show LDHB/Myc blots.
5. Fig 3A – Authors state the LDHB protein was “significantly” decreased at 24h post infection (line 141) – this decrease is not very apparent. Also, if the authors have done statistics over multiple western blots to determine significance, include the data in the figure.
6. Fig 4 –
4A - What software/resource was used to predict the LIR motif (line 162) – detail in methods.
4B – The IF images are too small to see. It is hard to see the canonical structures formed by the autophagosomes due to the small image size and it is hard to see the co-localization. Include quantification of the autophagosomes with HA-N overlap.

Minor Edits

1. Line 15 – Rephrase “to our interesting”
2. Line 123 – Change “clarify” to delineate.

Dear editor,

We deeply appreciate the time and effort you and the reviewers have spent in reviewing our manuscript titled “LDHB suppresses the PDCoV proliferation via targeting viral nucleocapsid protein for autophagic degradation” (ID: Spectrum02787-24). The comments are really thoughtful and helpful. Thus, we revised the manuscript, according to the comments carefully. Furthermore, all the modifications made in this revised manuscript are highlighted and labeled by yellow.

We hope you and the reviewers will be satisfied with the revisions for my manuscript. We are looking forward to your favorable reply.

Thank you very much.

Best regards.

Yours sincerely,

Wen Zhang

Reviewer 1:

In this paper, Wu et al. investigated the role of LDHB on PDCoV infection and replication. The authors found that the expression of LDHB was an important antiviral factor against PDCoV infection. Furthermore, the authors also demonstrated that the LDHB could interact with PDCoV N protein, and induce the autophagic degradation of PDCoV N protein. PDCoV N protein depending on its LIR interact with LC3 to degrade LDHB to escaping the host’s antiviral mechanism. There are some of issues that should be addressed by the authors.

Response: Thank you for reviewing our manuscript and for the constructive comments, which greatly helped us to improve the manuscript. And according to your suggestion, we have heavily revised our experiments. The manuscript was carefully revised and point-by-point response was listed below. We hope that your comments have been addressed accurately. The revised manuscript was marked with yellow color and the responses were presented in blue text.

(1) Lines 117-119: The authors point out “PDCoV N protein and LDHB co-localize at the mitochondria”. However, In Fig.2C the authors had proved that the LDHB and PDCoV N protein partially colocalized in the cytoplasm, but without any mitochondria marker presenting. Please provide the supporting data.

Response: Thanks for your suggestion. We are sorry for the ambiguous expression in the original manuscript, and we have stated clearly in the revised manuscript to avoid misunderstandings. The original sentences have changed by “These findings indicate that LDHB interact with PDCoV N protein in the cytoplasm to influence viral replication and cellular dynamics.” The changes could be found in line 120-122 of the revision.

(2) In the present study, authors only investigated the interaction between overexpressed LDHB and PDCoV N protein in Figure 2, the authors should verify the interaction between endogenous LDHB and PDCoV N protein during viral infection.

Response: Thanks for your suggestion. According to your suggestion, IP assay was performed using an antibody against PDCoV N in LLC-PK1 cells, both uninfected and infected with PDCoV. Subsequent blotting for the LDHB protein revealed that LDHB interacted with the PDCoV N protein specifically in the infected cells. These findings

have been added into the revised Figure 2 (Fig. 2B). In addition, we have made necessary corrections and extensions to the legend of the manuscript. We hope that the revised manuscript meets your expectations.

- (3) In the present study, authors only investigated the interaction between overexpressed GFP-LC3 and PDCoV N protein in Figure 5, the authors should verify the interaction between endogenous LC3 and PDCoV N protein during viral infection

Response: Thanks for your suggestion. We have conducted additional experiments in HEK-293T cells, where we transfected a plasmid encoding PDCoV N protein and performed co-immunoprecipitation assays with endogenous LC3. These results have been included in the revised manuscript as Figure 4D, with an expanded legend detailing the methods and findings. In addition, we have made necessary corrections and extensions to the legend of the manuscript. We hope that the revised manuscript meets your expectations.

- (4) Since your study focuses on the relationship between LDHB, N protein, and LC3, authors have only shown that N protein can induce the LC3-II increasing in the presence of LDHB. The interaction of LDHB and LC3 should be confirmed with or without PDCoV infection or PDCoV N protein expression, which help clarify the mechanism of LDHB and N protein interacting with LC3 for autophagic degradation. The results should be discussed in the part of Discussion.

Response: Thanks for your suggestion. In revision, we have included experiments that explicitly demonstrate the interaction between LDHB and LC3 under varying conditions of PDCoV N protein expression. We co-transfected Flag-LDHB, GFP-LC3, and HA-N protein plasmids to explore the effects of escalating N protein levels on LDHB stability and its interaction dynamics with LC3. The results demonstrate that increased expression of N protein markedly accelerates the degradation of LDHB in GFP-LC3-expressing cells (Fig. 4F). This reduction in LDHB availability results in decreased interaction signals between LDHB and LC3. These results suggest that PDCoV N promotes LDHB degradation through impacting autophagic flux by LC3 interaction mediated through its LIR motif. All the results could be found in the Line 176-184, and Figure 4F in the revision. In addition, we have made necessary corrections and extensions to the legend of the manuscript. We hope that the revised manuscript meets your expectations.

- (5) Line 15 and 214: The "To our interesting" should be corrected by "To our interests".

Response: Thank you for pointing out the errors in our manuscript. We have corrected it with your suggestion. It could be found in lines 15 and line 228 of the revision. We appreciate your attention to detail and assistance in improving the clarity and accuracy of our text.

- (6) Line 69-70: The cited content of the literature was described incorrectly. Please correct it.

Response: Thank you for bringing the incorrect literature citation to our attention. We have corrected the mistake. The changes could be found in Line 70 of the revision.

(7) Lines 168-170: The description of the LIR deletion mutant (N Δ LIR) is incomplete. The specific position and sequence of the LIR motif in the PDCoV N protein sequence needs to be added in the results and Fig.4A.

Response: Thanks for your suggestion. We have added the specific sequence of the LIR motif within the PDCoV N protein in the Results section. Furthermore, the sequence ASWFQV has been explicitly annotated in Figure 4A to clearly depict the LIR motif. In addition, we have made necessary corrections and extensions to the legend of the manuscript. We hope that the revised manuscript meets your expectations

(8) The materials and methods are too simple. Please provide more details. For example, the primer sequences for constructing plasmids should be added in Table 1, such as PDCoV N, LDHB and N- Δ LIR mutant.

Response: Thank you for your feedback regarding the level of detail in the Materials and Methods section of our manuscript. In the revision, we have added comprehensive details, including the primer sequences used for constructing the plasmids of PDCoV N, LDHB, and the N- Δ LIR mutant. These additions have been incorporated into Table 1. We hope that the revised manuscript meets your expectations

(9) Lines 406-432 : In the legends Fig.2 and Fig.3, the authors should add the details regarding the experimental conditions, such as the concentrations and treatment durations of the inhibitors.

Response: Thank you for your valuable feedback on the legends for Figures 2 and 3. We have updated both figures' legends to include detailed information about the experimental conditions, specifically noting the concentrations of inhibitors used and the duration of treatments. We hope that the revised manuscript meets your expectations.

(10) Overall, the English language should be improved by consulting native speaker or Author Writing Services.

Response: Thank you for reviewing our manuscript and for the constructive comments, which greatly helped us to improve the manuscript. And according to your suggestion, we have involved a native English professional teacher for language corrections. We really hope that our interpretation and the language in the revised manuscript will get your approval.

Reviewer 2:

In the manuscript titled "LDHB suppresses the PDCoV proliferation via targeting viral nucleocapsid protein for autophagic degradation," the authors investigate the role of LDHB in PDCoV replication. The study found that LDHB suppresses viral proliferation. The PDCoV-N protein was shown to interact with

LDHB. The authors also found an LIR motif within the N-protein. While the study has provided some compelling evidence showing LDHB affecting PDCoV levels, some major corrections to strengthen the paper are suggested below **Response: Thank you for your detailed feedback on our manuscript. We appreciate the opportunity to strengthen our paper and have corrected the manuscript according your suggestions. Each point has been carefully addressed to enhance the clarity and robustness of our findings. We look forward to your feedback on the revised manuscript and hope the changes meet your expectations. The revised manuscript was marked with yellow color and the responses were presented in blue text.**

- (1) Throughout the results, multiple cell lines have been used. Please indicate which cell line was used above/near each blot/IF/qPCR.

Response: Thank you for your suggestion. We have added specific annotations in the results and the figure legends to clearly indicate the cell line used for each experiment. Thanks again for your constructive feedback. We hope that the revised manuscript meets your expectations.

- (2) Beyond figure 1, none of the results have been validated in PDCoV infected cells.

Response: Thank you for your comment regarding the validation of our results in PDCoV-infected cells. To address your concern, I would like to clarify that beyond Figure 1, we have indeed conducted additional validations in the context of PDCoV infection:

- a) **In Figure 2B of the revision, we have added the interaction between PDCoV N protein and endogenous LDHB in PDCoV-infected cells.**
- b) **In Figure 2E of the revision, we validated the effects of inhibitor treatments in PDCoV-infected cells.**
- c) **In Figure 3A of the revision, we demonstrated the changes in LDHB expression in cells under PDCoV infection.**

These additional data points across multiple figures ensure that our findings are robust and consistently demonstrated in the context of viral infection. We hope this addresses your concerns and clarifies the scope of our validations. In addition, we have made necessary corrections and extensions to the legend of the manuscript. Thank you again for your attention to detail and for helping us improve the quality of our manuscript.

- (3) Fig 1E – In the blot confirming the efficacy of LDHB siRNA, at what time points were the samples collected?

Response: Thank you for your inquiry regarding the time points for sample collection in Figure 1E. I would like to confirm that the protein samples were collected 24 hours post-transfection with LDHB siRNA. This detail has been added to both the manuscript text and the legend for Figure 1E to ensure clarity about the experimental procedures.

Fig 1F – Changes in PDCoV N protein is not very apparent. Plot a bar graph with changes in PDCoV-N/GAPDH over multiple experiments and include other blots in supplemental data. It would be

helpful to confirm knockdown efficiency by LDHB western blots at the time points corresponding to 1F.

Response: Thank you for your valuable suggestion. In the revision, we have made necessary changes of the Figure 1 order and corrections and extensions to the legend of the manuscript. We have addressed your suggestion by incorporating a bar graph that quantifies the changes in the ratio of PDCoV N to GAPDH over multiple experiments directly in Figure 1G of the revision. Additionally, to confirm the knockdown efficiency of LDHB, we have included quantitative results of LDHB expression levels in the same samples in Figure 1F of the revision. These enhancements provide a clearer visual representation of the data and confirm the efficacy of LDHB knockdown at the corresponding time points. We hope that these amendments improve the clarity and rigor of our findings.

- (4) Fig 2 – The data would be compelling if authors could replicate the experiment in 2E (using the various autophagy inhibitors) on PDCoV infected LLC-PK1 and measure viral replication to further validate the role of LDHB in an autophagy-mediated mechanism.

Response: Thank you for your suggestion. According your suggestion, we performed the supplemental assay. We also collected a consistently results and added it as Fig. 2G of the revision. This addition strengthens our findings by directly linking LDHB's function to viral replication in the context of autophagy inhibition. In addition, we have made necessary corrections and extensions to the legend of the manuscript. We hope that the revised manuscript meets your expectations.

2C - The IF images are too small to see.

Response: Thank you for pointing out the issue with the size of the immunofluorescence images in Figure 2C. In revision, we have enlarged the co-localization sections of the images to enhance visibility and detail. We hope that this adjustment improves the clarity of the results presented. We hope that the revised manuscript meets your expectations.

2F – Include images with darker/longer exposure to see the LC3-I band. And, include LDHB/Flag blot(s).

Response: Thank you for your valuable feedback regarding the visibility of the LC3-I band and the inclusion of LDHB/Flag blots. In response to your comments, we have updated the images with darker and longer exposures, ensuring that both LC3-II and LC3-I bands are clearly visible. Additionally, we have included a bar graph illustrating the ratio of LC3-II to LC3-I across multiple experiments to provide a more comprehensive analysis. We have also added the protein blot images for Flag-LDHB to further substantiate our findings. In addition, we have made necessary corrections and extensions to the legend of the manuscript. We hope that the revised manuscript meets your expectations.

2G - Show LDHB/Myc blots.

Response: Thank you for your suggestion to include Myc-LDHB blots in our presentation of results. We have added immunoblot images showing Myc-LDHB to the relevant figure in the revision. We appreciate your attention to detail and believe that this enhancement strengthens the supporting evidence for our study. In addition, we have made necessary corrections and extensions to the legend of the manuscript. We hope that the revised manuscript meets your expectations.

(5) Fig 3A – Authors state the LDHB protein was “significantly” decreased at 24h post infection

(line 141) – this decrease is not very apparent. Also, if the authors have done statistics over multiple western blots to determine significance, include the data in the figure.

Response: Thank you for pointing out the issue with the description of LDHB protein levels in Figure 3A. We have revised the manuscript to accurately state that the decrease in LDHB protein at 24 hours post-infection is not significant, aligning the text with the observed data. The changes could be found in Line 148-150 of the revision. I appreciate your attention to this detail, which has helped clarify the presentation of our results. We hope that the revised manuscript meets your expectations.

(6) Fig 4 –

4A - What software/resource was used to predict the LIR motif (line 162) – detail in methods.

Response: Thank you for your inquiry about the methods used to predict the LIR motif. We used the iLIR@Viral tool for this purpose. The website URL for this resource has been included in the legend of Figure 4A to provide full transparency about the tools employed in our analysis. We hope that the revised manuscript meets your expectations.

4B – The IF images are too small to see. It is hard to see the canonical structures formed by the autophagosomes due to the small image size and it is hard to see the co-localization. Include quantification of the autophagosomes with HA-N overlap.

Response: Thank you for your feedback regarding the size of the immunofluorescence images in Figure 4B. We have included enlarged images to improve visibility of the structures formed by autophagosomes and to better demonstrate their presence. Additionally, we have added another set of laser confocal image, which are from the GFP-LC3 with or without PDCoV N expression in LLC-PK1 cells. The results showed that PDCoV N protein promotes the formation of LC3 aggregates, and presents a colocalization signal in the cytoplasm. These images could be found in Fig.4B (down) of the revision. It is important to clarify that our primary focus in these experiments was aimed at providing proof of concept rather than quantitative data. We hope that the inclusion of enlarged and added images could help to address your concerns and better convey the intentions of our study. In addition, we have made necessary corrections and

extensions to the legend of the manuscript. We hope that the revised manuscript meets your expectations.

(7) Line 15 – Rephrase “to our interesting”

Response: Thank you for pointing out the phrasing issue in Line 15 and Line 228. We have corrected the phrase "to our interesting" to "To our interests" in the revision. We hope that the revised manuscript meets your expectations.

(8) Line 123 – Change “clarify” to delineate.

Response: Thank you for your suggestion. We have changed "clarify" to "delineate" as recommended to enhance the precision of the language used. The change could be found in Line 127 of the revision.

Re: Spectrum02787-24R1 (LDHB suppresses the PDCoV proliferation via targeting viral nucleocapsid protein for autophagic degradation)

Dear Prof. Wen Zhang:

Your manuscript has been accepted, and I am forwarding it to the ASM production staff for publication. Your paper will first be checked to make sure all elements meet the technical requirements. ASM staff will contact you if anything needs to be revised before copyediting and production can begin. Otherwise, you will be notified when your proofs are ready to be viewed.

Sincerely,
Leiliang Zhang
Editor
Microbiology Spectrum